# TRIP13 Participates in Immediate-Early Sensing of DNA Strand Breaks and ATM Signaling Amplification through MRE11

**DOI:** 10.3390/cells11244095

**Published:** 2022-12-16

**Authors:** Hyeongsun Jeong, Minwoo Wie, In-Joon Baek, Gyuwon Sohn, Si-Hyeon Um, Seon-Gyeong Lee, Yuri Seo, Jaesun Ra, Eun A Lee, Shinseog Kim, Byung Gyu Kim, Rajashree A. Deshpande, Tanya T. Paull, Joo Seok Han, Taejoon Kwon, Kyungjae Myung

**Affiliations:** 1Center for Genomic Integrity, Institute for Basic Science, Ulsan 44919, Republic of Korea; 2Department of Biomedical Engineering, College of Information and Biotechnology, Ulsan National Institute of Science and Technology, Ulsan 44919, Republic of Korea; 3Department of Biological Sciences, College of Information and Biotechnology, Ulsan National Institute of Science and Technology, Ulsan 44919, Republic of Korea; 4Graduate School of Analytical Science and Technology, Chungnam National University, Daejeon 34134, Republic of Korea; 5Department of Molecular Biosciences, The University of Texas at Austin, Austin, TX 78712, USA; 6Encoded Therapeutics, Inc., San Francisco, CA 94080, USA; 7Neuracle Genetics, Seoul 02841, Republic of Korea

**Keywords:** HORMA domain, DNA damage response, TRIP13, MRN complex

## Abstract

Thyroid hormone receptor-interacting protein 13 (TRIP13) participates in various regulatory steps related to the cell cycle, such as the mitotic spindle assembly checkpoint and meiotic recombination, possibly by interacting with members of the HORMA domain protein family. Recently, it was reported that TRIP13 could regulate the choice of the DNA repair pathway, i.e., homologous recombination (HR) or nonhomologous end-joining (NHEJ). However, TRIP13 is recruited to DNA damage sites within a few seconds after damage and may therefore have another function in DNA repair other than regulation of the pathway choice. Furthermore, the depletion of TRIP13 inhibited both HR and NHEJ, suggesting that TRIP13 plays other roles besides regulation of choice between HR and NHEJ. To explore the unidentified functions of TRIP13 in the DNA damage response, we investigated its genome-wide interaction partners in the context of DNA damage using quantitative proteomics with proximity labeling. We identified MRE11 as a novel interacting partner of TRIP13. TRIP13 controlled the recruitment of MDC1 to DNA damage sites by regulating the interaction between MDC1 and the MRN complex. Consistently, TRIP13 was involved in ATM signaling amplification. Our study provides new insight into the function of TRIP13 in immediate-early DNA damage sensing and ATM signaling activation.

## 1. Introduction

The HORMA domain was initially identified as a highly conserved peptide sequence in three yeast proteins that maintain genome stability in different processes: Hop1 (HORMAD1 in human) in meiotic recombination and chromosome segregation, Rev7 (MAD2L2) in the recombination choice during DNA double-strand break (DSB) repair, and Mad2 (MAD2L1) in the spindle assembly checkpoint [1,2,3]. Recent studies identified proteins with this domain that function as signal mediators, such as MAD2L1BP (also known as p31(comet)) for spindle assembly complex signaling, and ATG13 and ATG101 for autophagy signaling [2]. These proteins form physical protein–protein interactions via their HORMA domain to regulate signaling pathways; therefore, it is essential to understand their protein interaction partners.

Thyroid hormone receptor-interacting protein 13 (TRIP13, called Pch2 in yeast), a typical AAA+ ATPase, is a binding partner of members of the HORMA domain protein family in multiple contexts. In meiotic G2/prophase, TRIP13 is enriched in the nucleolus and actively involved in removing HORMAD1 from chromosomes in yeast and mammals [4]. Additionally, TRIP13, together with the MAD2-binding protein MAD2L1BP, promotes inactivation of the spindle assembly checkpoint by disassembling the mitotic checkpoint complex [5]. This checkpoint system ensures the accuracy of chromosome segregation by delaying the anaphase until the correct bipolar attachment of chromatids to the mitotic spindle is achieved. Due to its essential role in genome stability, the mutation of TRIP13 is highly associated with cancers, such as Wilms tumor [6], glioblastoma [7], and head and neck cancer [8].

Recently, it was reported that TRIP13 plays a role in DSB repair by determining the choice between the homology-directed repair (HDR) pathway and the nonhomologous end-joining (NHEJ) pathway [3,9]. The HORMA domain-containing protein MAD2L2 (Rev7) is a subunit of the Shieldin complex that protects DSBs and promotes DSB end-joining. It also participates in DNA translesion synthesis as a member of the Pol-ζ complex. The interaction between TRIP13 and MAD2L2 (Rev7) was previously reported using a high-throughput yeast two-hybrid screen [10]. In addition, two studies reported that this interaction modulates the conformation of MAD2L2 (Rev7), which determines the choice of the DNA repair pathway [3,9].

Although TRIP13 interacts with proteins with a HORMA domain, it has also been reported to interact with proteins without this domain [11,12]. For example, the TRIP13 ortholog in *S. cerevisiae*, Pch2, interacts with Xrs2 (the ortholog of human NBS1) based on a yeast two-hybrid assay [11]. Disruption of the interaction between Pch2 and Xrs2 causes checkpoint and recombination defects similar to those found in *Pch2-*knockout cells [11]. Both Pch2 and Xrs2 are conserved in mammals, but the interaction of their orthologs, TRIP13 and NBS1 in humans, has not been reported.

Here, we identified 279 proteins as interaction partners of TRIP13 based on a proximity labeling-based quantitative proteomics experiment. Many of these proteins were related to the DNA damage response, such as MRE11 of the MRN complex. This led us to speculate that TRIP13 and other proteins with a HORMA domain play a role in the DNA damage response. We revealed that TRIP13 participates in immediate-early DNA damage sensing and regulates ATM signaling independently of the MRN complex. 

## 2. Materials and Methods

### 2.1. Cell Culture

ER-*AsiS*I U-2-OS cells were kindly gifted by Dr. Tanya Paull (The University of Texas at Austin, Austin, TX, USA) [13]. DLD-1 cells with an auxin-inducible degron in both TRIP13 alleles (DLD1-TRIP13-AID) were previously reported [14]. DLD-1 and U-2-OS cells were grown in Dulbecco’s Modified Eagle’s Medium (11965092; Thermo Fisher Scientific, Waltham, MA, USA) supplemented with 10% fetal bovine serum (FBS; TMS-013-BKR; Merck, Rahway, NJ, USA) and 1% antibiotic-antimycotic solution (15240112; Invitrogen, Waltham, MA, USA). U2OS cells carrying DR-GFP (Homologous recombination), SA-GFP (Single strand annealing), EJ2-GFP (Micro-homology mediated end joining), or EJ5-GFP (non-homologous end joining) reporters were grown in DMEM containing 10% fetal bovine serum, 1% penicillin/streptomycin (penicillin 10,000 units/mL, streptomycin 10,000 µg/mL, Gibco^®^, Thermo Fisher Scientific, Waltham, MA, USA) and 2 µg/mL Puromycin.

### 2.2. Neutral COMET Assay

The COMET assay was performed using a CometAssay^®^ kit (Trevigen, Gaithersburg, MA, USA) according to the manufacturer’s instructions. In brief, each cell suspension (1 × 10^5^/mL) was mixed with COMET LM Agarose at 37 °C and the mixture was spread on a 2 well COMET slide^TM^ (Trevigen). After solidification of the agarose for 10 min at 4 °C, the slide was immersed in a lysis solution (Trevigen) overnight at 4 °C. The slide was immersed in 1× TBE buffer for 15 min at 4 °C. The slide was electrophoresis in 850 mL 1 × TBE buffer for 40 min at 4 °C. The slide was washed twice in distilled water and immerse in 70% EtOH for 5 min. After dry the slide at 37 °C, the slide was stained by SYBR Gold for 30 min at room temperature. Images were acquired with a fluorescence microscope (BX53; Olympus, Tokyo, Japan), and the tail moment was calculated using CometScore software version 2.0.

### 2.3. Sister Chromatid Exchange (SCE) Assay

Cells were grown in a medium containing BrdU at a final concentration of 25 µg/mL for 48 h, then 0.2 µg/mL of colcemid was added during the final four hours. Metaphase cells were collected by trypsinization, and swelled in 0.075 M KCl for 15 min at 37 °C and fixed twice with a 3:2 mixture of methanol and acetic acid. On glass microscope slides, cells were placed and stained with a 5% Giemsa solution.

### 2.4. Plasmids

Bio-ID2 (from #74224, Addgene, Watertown, MA, USA) was inserted into the previously generated pcDNA5-Myc-TRIP13 plasmid [14]. The pcDNA5-2xStrep-TRIP13 plasmid was generated by inserting the 2xStrep tag from the pDSG-IBA103 vector (5-5105-001; IBA, Göttingen, Germany) into the pcDNA5-Myc-TRIP13 vector. Each NBS1 domain deletion series (FHA for 1–109 aa, BRCT for 110–327 aa, and C-term for 328–754 aa) was replaced by full-length *NBS1* from the pcDNA5-Myc-BioID2-NBS1 vector by Gibson assembly cloning. For microirradiation, the pENTR-GFP-MDC1 plasmid was purchased from Addgene (#26284). Using Gibson assembly cloning, the designated *MRE11* fragment was inserted into the pcDNA5 3xFLAG vector for in vitro pulldown assay. For the MRN nuclease activity assay, full-length TRIP13 was purchased from OriGene Technologies (Rockville, MA, USA), and mutated forms (K185A and E253Q) were generated by site-directed mutagenesis. Wild-type TRIP13 and its two variant forms (K185A and E253Q) were inserted into the pET-His6-Sumo-TEV-LIC cloning vector (#29659, Addgene). *I-Sce*I (pCAGGS-I-*Sce*I, called pCBASce), empty vector (pCAGGS-BSKX), and dsRED vector were prepared as previously described [15,16,17].

### 2.5. Antibodies

A list of antibodies used in this study is provided in Table 1.

### 2.6. Regulation of TRIP13 Expression and MRE11 Exonuclease Activity

DLD1-TRIP13-AID cells were treated with 500 µM IAA (1003530010; Merck Millipore, Burlington, MA, USA) for 14 h to deplete TRIP13 as previously described [14]. Treatment with 100 µM mirin (J67462.MA; Thermo Fisher Scientific, Waltham, MA, USA) was also performed to inhibit MRE11 exonuclease activity.

### 2.7. DSB Induction

DSBs were induced in cultured cells, releasing one hour after exposure to 5 or 10 Gy IR using SkyScan 1176 (SKYSCAN, Kontich, Belgium) or 1 µM of CPT treatment for one hour. For microirradiation, cells in fresh medium were pretreated with Hoechst 33342 (62249, Thermo Fisher Scientific) for 5 min at 37 °C and then irradiated in a region of interest (ROI) with a 405 nm laser for ten iterations using a LSM 880 confocal microscope (Carl Zeiss AG, Oberkochen, Germany). Time-lapse images were acquired every 5 sec after microirradiation with a LSM 880 confocal microscope. Intensities in the ROI were analyzed using ZEN blue software (Version 2.3, Carl Zeiss AG). For comet assay, we applied one hour release after 30 Gy irradiation, 10 mM HU treatment for 4 h and 25 µM CPT treatment for one hour for damage induction, respectively.

### 2.8. Immunofluorescence Imaging

Cells were treated with 1 mL ice-cold permeabilization solution (CSK buffer; 10 mM PIPES, 100 mM NaCl, 300 mM sucrose, 3 mM MgCl_2_, and 1 mM EGTA containing 0.5% Triton X-100) for 10 min. Thereafter, cells were washed twice with 1 mL ice-cold phosphate-buffered saline (PBS) and then fixed with 500 µL PBS containing 4% paraformaldehyde for 15 min at room temperature. Cells were further incubated with 1 mL of 100% methanol chilled to −20 °C for 10 min, washed twice with 1 mL PBS, and incubated with 500 µL blocking solution (PBS containing 10% FBS) for 30 min. The blocking solution was removed, and a primary antibody was treated for one hour. Cells were washed with PBS containing 0.05% Triton X-100 for 5 min, incubated with a secondary antibody for 30 min at room temperature in the dark, and washed thrice with PBS containing 0.05% Triton X-100 for 5 min each time. An additional washing step with PBS and distilled water was performed, and the surrounding chamber was removed from the slide. A drop of mounting reagent (H-1200; Vector Laboratories) was applied to each sample, and the slide was covered with a cover glass. The slides were dried and completely sealed. Protein foci were detected and visualized using an LSM 880 confocal microscope and ZEN software, counting the number of foci overlapped with the DAPI. Statistical analysis was performed with Prism5 software (Version 5.01; GraphPad, San Diego, CA, USA).

### 2.9. Proteomics Experiments with SILAC

293AD cells (Cell BioLabs, Inc., San Diego, CA, USA) cultured in medium containing light isotope-labeled lysine and arginine (88429 and 89989, Thermo Fisher Scientific) and medium containing heavy isotope-labeled lysine and arginine (89990 and 88209, Thermo Fisher Scientific) were transfected with Myc-BioID2-TRIP13 and Myc-BioID2 plasmids, respectively. Cell lysates were mixed at a 1:1 ratio and processed by in-gel digestion, as previously described [18]. A total of 20 mg protein was reduced with 25 mM DTT, alkylated with 25 mM iodoacetamide, and digested with trypsin at 37 °C for 12 h. The solution was acidified with 0.1% trifluoracetic acid and desalted on C18 tips (87784, Thermo Fisher Scientific). Samples were analyzed using Orbitrap Fusion Lumos (Thermo Fisher Scientific). Proteins were identified using Proteome Discoverer (Thermo Fisher Scientific) and Scaffold software (Version 4.11.0; Proteome Software, Portland, OR, USA).

### 2.10. Western Blot Analysis

Cells were lysed in RIPA buffer (150 mM NaCl, 1% NP-40 or Triton X-100, 0.5% sodium deoxycholate, 0.1% SDS, 50 mM Tris (pH 8.0), 10 mM NaF, 1 mM Na_3_VO_4_, and protease inhibitors (Roche, Basel, Switzerland)). Lysates were heated thrice for 5 min at 95 °C. Proteins in prepared samples were separated by SDS-PAGE, transferred to nitrocellulose membranes for Western blot analysis, and detected using the antibodies listed in Table 1. Images were quantitatively analyzed using an Odyssey Imaging System (Li-COR Biosciences, Lincoln, NE, USA).

### 2.11. In Vitro Pulldown Assay

Purified proteins were incubated at 4 °C for 1 h in buffer (50 mM Tris-HCl, 50 mM NaCl, 1 mM DTT, 2 mM MgCl_2_, and 0.1% Triton X-100) supplemented with 10 μM ATPγS (11162306001, Roche). MagStrep type 3 XT beads or 3xFLAG agarose beads were incubated with purified proteins for 1 h at 4 °C. Samples were eluted in BXT buffer or Tris-buffered saline (50 mM Tris-HCl pH 7.5 containing 150 mM NaCl) containing 100 μg/mL 3xFLAG peptide.

### 2.12. Immunoprecipitation

Cells were lysed by incubating cells in ice-cold buffer X (100 mM Tris-HCl, 250 mM NaCl, 1 mM EDTA, 1% NP-40) with Halt™ Protease and Phosphatase Inhibitor Single-Use Cocktail (100×) and Benzonase^®^ nuclease (250 unit/µL, Enzynomics, Daejeon, Republic of Korea) for 1 h with rotation at 4 °C. The lysate was centrifuged at 13,000 rpm for 15 min. Protein amounts were quantified with Pierce™ Detergent Compatible Bradford Assay Kit (Thermo Fisher Scientific). Flag beads (ANTI-FLAG^®^ M2 Affinity Gel, Sigma-Aldrich, Saint Louis, MO, USA) were washed 3 times with ice-cold buffer X. The lysate was incubated with 28.6 µL bead slurry for 2 h with rotation at 4 °C. The beads were washed with ice-cold buffer X for 5 min, with rotation three times in 5 min intervals. After washing, beads were incubated overnight in 35 µL elution buffer (167 µg/mL FLAG peptide, 1X Halt™ PI cocktail, buffer X) at 4 °C.

### 2.13. Colony Formation Assay

A total of 5 × 10^5^ cells were seeded in a 35 mm dish, grown overnight, and treated with the designated chemicals and IR. Then, 500 cells were spread in a 35 mm dish, incubated for 14 days, washed with 5 mL PBS, and stained with 0.5 mL of 1% methylene blue prepared in 70% ethanol for 5 min at room temperature. After removing methylene blue, stained cells were washed with distilled water and PBS, three times each. Dishes were dried for 24 h at room temperature, and images of colonies were acquired. The number of colonies was counted using ImageJ (Version 1.51) [19]. Statistical analyses were performed using Prism5 software (Version 5.01, GraphPad).

### 2.14. I-SceI-Induced Assay

U2OS cells stably expressing DR-GFP, SA-GFP, EJ2-GFP, or EJ5-GFP reporter were plated on a 12-well plate at 1 × 10^5^ cells per well density. Cells were co-transfected cells with 0.5 µg of either I-*Sce*I expression vector or empty vector with 0.1 µg of dsRED vector (used as a transfection control) using 0.1 mL Opti-mem containing Lipofectamine 3000 (Invitrogen). The percentage of GFP+ cells was analyzed by the Becton Dickinson FACSVerse flow cytometer (BD Biosciences, San Jose, CA, USA). The DNA repair frequencies were determined as previously described [15]. The experiment was repeated twice, and the two-tailed unpaired T-test was performed to determine significance.

### 2.15. DNA End Resection Assay

ER-*AsiS*I U-2-OS cells were trypsinized, harvested, and resuspended in PBS (BE17-517Q; Lonza, Basel, Switzerland) containing 0.6% low-gelling point agarose (1613111; Bio-Rad, Hercules, CA, USA) at 37 °C. Then, 50 μL of the cell suspension was dropped on a piece of parafilm to generate a solidified agar ball, which was transferred to a 1.5 mL tube. The agar ball was treated with 1 mL ESP buffer (0.5 M EDTA, 2% N-lauroylsarcosine, 1 mg/mL proteinase K, and 1 mM CaCl_2_, pH 8.0) for 20 h at 16 °C with rotation, followed by 1 mL HS buffer (1.85 M NaCl, 0.15 M KCl, 5 mM MgCl_2_, 2 mM EDTA, 4 mM Tris, and 0.5% Triton X-100, pH 7.5) for 20 h at 16 °C with rotation. After six washes with 1 mL phosphate buffer (8 mM Na_2_HPO_4_, 1.5 mM KH_2_PO_4_, 133 mM KCl, and 0.8 mM MgCl_2_, pH 7.4) for 1 h each time at 4 °C with rotation, the agar ball was melted by placing the tube in a 70 °C heat block for 10 min. The melted sample was diluted 15-fold with 70 °C distilled water, mixed with 10× NEB restriction enzyme buffer, and stored at 4 °C.

The level of resection adjacent to specific DSBs was measured by qPCR. The sequences of qPCR primers and probes are provided in Table 2. In total, 20 μL of genomic DNA was digested with 20 units of restriction enzymes (*BsrGI* and *Hind*III-HF; New England Biolabs, Ipswich, MA, USA) or mock digested at 37 °C overnight. Then, 3 μL of digested or mock digested samples was used as templates in a qPCR reaction with a total volume of 25 μL and containing 12.5 μL of 2xTaqMan Universal PCR Master Mix (4304437, Thermo Fisher), 0.5 mM of each primer, and 0.2 mM of the probe using a ViiATM 7 Real-Time PCR System (Thermo Fisher). The percentage of ssDNA (ssDNA%) generated by resection was determined with the following equation as previously described [13]: ssDNA% = 1/(2^(ΔCt − 1)^ + 0.5) × 100. The percentage of ssDNA in each sample was analyzed using Prism5 software (Version 5.01, GraphPad).

### 2.16. MRN Endonuclease Activity Assay

Wild-type TRIP13 and its variant forms (K185A and E253Q) were purified as previously described [14]. A previously reported protocol was used for the activity assay, including purification of members of the MRN complex and other associated proteins [20].

## 3. Results

### 3.1. TRIP13 Participates in the DNA Damage Response

Previous studies claimed that TRIP13 regulates the choice of the DSB repair pathway through MAD2L2 (Rev7) [3,9]. However, these studies only investigated the impact of TRIP13 on the HDR pathway. Another study revealed that the role of TRIP13 in NHEJ through immunoglobulin class switch recombination (CSR) efficiency increases under TRIP13 depletion [21]. However, the overexpression of TRIP13 in TRF2ts MEFs did not affect the number of chromosome end-to-end fusions caused by telomere uncapping and the reduced number of chromosomal fusions by TRIP13 knockdown was lesser than the extent of MAD2L2 knockdown. To investigate whether TRIP13 has other roles in DSB repair pathways, we performed an I-*Sce*I-based GFP-reporter assay upon TRIP13 knockdown [22]. The depletion of TRIP13 affected the HDR and single-strand annealing (SSA) pathways (Figure 1A), consistent with previous reports [3,9]. However, a significant decrease in the NHEJ and alternative NHEJ (alt-NHEJ) pathways were also observed, although they are not large (Figure 1A). These results suggest that TRIP13 is important for DSB repair, especially that requiring DNA end resection.

Next, we performed a laser microirradiation experiment to investigate when TRIP13 is recruited to DSBs. Proteins recruited to DNA lesions could be damage sensors, chromatin remodelers/signal amplifiers, and lesion repair choice/repair proteins. Their recruitment kinetics differ depending on their functions [23]. If TRIP13 is only involved in the DSB pathway choice, it may not necessarily be recruited to lesions at the early stage for damage sensing/signal amplification. TRIP13 binds to the substrate with ATP and releases the substrate after hydrolyzing ATP to change the structures of its interaction partners; therefore, we used ATPγS, a non-hydrolysable ATP analog, to stably maintain the interactions between TRIP13 and its partners. In the absence of ATPγS, the recruitment of TRIP13 on the damaged site was not clear. The addition of ATPγS clearly shows that TRIP13 was immediately recruited to the DNA damage site when a DSB was induced (Figure 1B). Therefore, we speculated that TRIP13 may be involved in DNA damage sensing and signal amplification at the immediate-early stage.

To more precisely control TRIP13 expression, we utilized the auxin-inducible degron (AID) system, which allows the rapid and effective degradation of target proteins [24]. TRIP13 depletion using the AID system sensitized cells to the induction of DSBs by ionizing radiation (IR) or hydroxyurea (HU) treatment (Figure 1C). Furthermore, DNA damage sensitization (IR, HU, CPT) under TRIP13 degradation was also observed in neutral comet assay (Figure 1D). These results confirmed that TRIP13 is also involved in the immediate-early response to DSBs.

### 3.2. MRE11 Is a Novel Interaction Partner of TRIP13, but this Interaction Is Independent of the Conformation and Nuclease Activity of the MRN Complex

The interactions of TRIP13 with its partners are highly dynamic and thus difficult to identify using the conventional immunoprecipitation approach. Therefore, we performed quantitative proteomics with proximity labeling to identify proteins located near TRIP13, including its interaction partners. First, we established a U-2-OS cell line expressing BioID2-tagged TRIP13 [25] and then metabolically labeled these cells for stable isotope labeling using amino acids in cell culture (SILAC) [26,27]. We used U-2-OS cells expressing BioID2 only as a negative control. Finally, we captured biotin-labeled proteins enriched in BioID2-tagged TRIP13-expressing cells, which may be located near TRIP13 in vivo. TRIP13 is rapidly recruited to DNA damage sites, and therefore we reasoned that the identification of its intrinsic interaction partners in the absence of DNA damage would be informative to understand its function. Consequently, we performed this experiment without inducing DNA damage.

TRIP13 was the top-ranked protein among enriched proteins. Among the 279 identified proteins (the complete list of enriched proteins is provided in Appendix A), we focused on proteins related to the DNA repair pathway. MRE11, a member of the MRN complex, was one of the top candidates for a TRIP13 interaction partner (Figure 2A). Using the proteins (p31comet, MAD2, and CDC20) known to interact with TRIP13, the proximal localization using BioID2-TRIP13 was confirmed by Western blotting following biotin pulldown (Figure 2B). Furthermore, the direct interaction between MRE11 and TRIP13 was confirmed by affinity precipitation using 2xStrep-tagged TRIP13, which interacted with both endogenous MRE11 (Figure 2C) and purified 3xFLAG-tagged MRE11 (Figure 2D).

Previously, it was reported that NBS1, another member of the MRN complex, is a candidate interaction partner of TRIP13 [11]. Therefore, we tested the interaction between TRIP13 and other components of the MRN complex upon induction of DNA strand breaks by camptothecin (CPT) treatment (Figure 2E), which inhibits topoisomerase I during DNA replication. The association between TRIP13 and the MRN complex was detected in the presence and absence of DNA damage. Therefore, we speculated that TRIP13 might have a role in MRE11 regulation, even in the absence of DNA damage. In addition to MRE11 and RAD50, which are components of the MRN complex, ATM and CtIP, which are MRN complex cofactors, were labeled by BioID2 fused with TRIP13. However, we did not detect any NBS1 peptide; therefore, BioID2 tagged with TRIP13 may not be labeled NBS1. Therefore, we concluded that TRIP13 interacts with MRE11, but not with NBS1, in the MRN complex.

To identify which domain in MRE11 is responsible for TRIP13 interaction, 3xFLAG-TRIP13 IP was performed with overexpressing Myc-MRE11 domain deletion series (Figure 2F). We divided the MRE11 domain by the nuclease domain, capping domain and C-terminal domain that contains the glycine–arginine domain (GAR) and DNA binding domain. Capping domain deletion or C-terminal domain deletion did not seem to affect the interaction with TRIP13. So, we speculate that the nuclease domain is responsible for the TRIP13 interaction, but further studies would be needed.

Next, we investigated whether TRIP13 regulates the stability of the MRN complex, which is essential for its physical interactions. Surprisingly, the associations between Mre11 and other components of the MRN complex were unchanged regardless of the presence or absence of irradiation and TRIP13 (Figure 2G). In addition, an MRN endonuclease in vitro assay with TRIP13 wild-type, KA mutant (ATP-binding mutant), and EQ mutant (ATP hydrolysis mutant) showed that TRIP13 did not affect the endonuclease activity of the MRN complex (Figure 2H). These results suggest that TRIP13 participates in the DNA damage response independently of the endo/exonuclease activity of the MRN complex.

### 3.3. TRIP13 Depletion Inhibits the Physical Interaction of MDC1 with MRE11 and Reduces the Recruitment of MDC1 to DNA Damage Sites

DSBs are recognized by the MRN complex, which binds to DSB sites directly [28,29,30]. The MRN complex then recruits and activates ATM kinase, which phosphorylates histone H2AX (γH2AX) at DSB sites, which in turn provides a binding site for MDC1 [31]. MDC1 at DSB sites recruits more MRN-ATM complex to areas surrounding DSB sites, leading to γH2AX-MDC1 spreading and amplifying ATM signaling on adjacent chromatin [31]. We hypothesized that TRIP13 functions in the recruitment of MDC1 to DNA damage sites.

The chromatin-bound fraction of MDC1 was increased by IR-induced DNA damage and decreased by indole-3-acetic acid (IAA)-induced TRIP13 degradation (Figure 3A). MDC1 binds to the MRN complex directly through phosphorylated SDTD motifs [28,32,33,34]; therefore, we speculated that this interaction is regulated by TRIP13. The direct interaction between MDC1 and the MRE11 complex was validated by the immunoprecipitation of MRE11 (Figure 3B). The interaction of MDC1 and MRE11 was maintained regardless of TRIP13 without DNA damage but was significantly reduced in the absence of TRIP13 upon DNA damage (Figure 3B).

To investigate how TRIP13 depletion affects chromatin loading of MDC1, we analyzed the formation of γH2AX and MDC1 foci (Figure 3C). The number of MDC1 foci was significantly reduced upon TRIP13 depletion alone. To validate the role of TRIP13 in recruitment of MDC1 to DNA damage sites, we performed laser micro-irradiation of GFP-tagged MDC1-expressing cells. When TRIP13 was knocked down, recruitment of MDC1 was not only reduced but also delayed (Figure 3E). MDC1 binds directly to the C-terminal region of γH2AX [31,35], and the presence of MDC1 foci depends on γH2AX foci [36]. Surprisingly, depletion of TRIP13 did not affect γH2AX foci (Figure 3C). Similarly, 100 µM mirin treatment, which inhibits MRE11 nuclease activity, reduced the number of MDC1 foci but not that of γH2AX foci. However, the combination of mirin treatment and TRIP13 depletion inhibited the formation of both MDC1 and γH2AX foci (Figure 3C). Our results suggest that both mirin and TRIP13 regulate γH2AX phosphorylation, whereas the recruitment of MDC1 to DNA damage sites is dependent on TRIP13. It is not clear whether TRIP13 regulates the ATM signaling pathway independently of mirin because it was reported that mirin treatment at a high dose (over 500 µM) could block ATM activity [37]. In contrast to MDC1, we could not observe a significant decrease in phosphorylated ATM foci formation with TRIP13 depletion under IR irradiation (Figure 3D). Since mirin treatment alone suppressed ATM phosphorylation sufficiently, we could not evaluate the effect of TRIP13 depletion under MRE11 inhibition with mirin treatment. Collectively, TRIP13 promotes the recruitment of MDC1 to DNA damage sites by regulating the physical interaction between the MRN complex and MDC1.

### 3.4. TRIP13 Enhances DNA Resection and Homologous Recombination by Regulating ATM Downstream Signaling

TRIP13 regulates MDC1 recruitment to DNA damage sites through the MRN complex; therefore, we investigated which signaling pathways downstream of DSB sensing are controlled by these interactions. The phosphorylated forms of ATM kinase substrates, such as ATM, NBS1, KAP1, and RPA32(S4/8), were not fully inhibited on either mirin treatment or IAA-induced TRIP13 depletion but were downregulated upon TRIP13 depletion with mirin (Figure 4A). Interestingly, TRIP13 depletion did not affect RPA32(S33) phosphorylated by ATR kinase. Although inhibition of the MRN complex by mirin treatment affects both ATM and ATR pathways, TRIP13 suppression only affects the ATM pathway. Additionally, TRIP13 depletion and MRN complex inhibition by mirin treatment elicited an additive inhibitory effect on the phosphorylation of ATM downstream proteins, suggesting that TRIP13 regulates DSB signaling proteins in a mirin-independent manner.

TRIP13 depletion reduced the phosphorylation of RPA32(S4/8). Therefore, we performed a DNA resection assay using the ER-*AsiS*I system to determine the effect of TRIP13 knockdown on DNA resection (Figure 4B). DNA resection was quantified at 335, 1618, and 3500 bp from the DSB by ER-*Asi*SI [38]. Long-range resection (3500 bp) from the DSB did not occur, even in the control group in our experimental schedule. TRIP13 depletion or mirin treatment reduced short-range (335 bp) and intermediate-range (1618 bp) resection. Similar to the effects on ATM signaling (Figure 3C and Figure 4A), combinatorial treatment with TRIP13-targeting siRNA (Table 3) and mirin additively reduced both short- and intermediate-range resection.

The sister chromatid exchange (SCE) frequency is a representative index of HR [39,40,41]. We investigated the role of TRIP13 in SCE. As expected, TRIP13 depletion or mirin treatment reduced the SCE frequency, and their effects were additive (Figure 4C). We also investigated the effect of TRIP13 on the HDR and SSA frequencies using the GFP-reporter system. Both TRIP13 depletion and mirin treatment blocked the HDR pathway, similar to the effect of RAD51 knockdown (Figure 4D). Furthermore, both TRIP13 depletion and mirin treatment inhibited the SSA pathway, similar to the effect of RAD52 knockdown (Figure 4E). These results indicate that TRIP13 is involved in ATM signaling and the HDR pathway via its direct interaction with MRE11, which regulates interactions between MDC1 and the MRN complex.

## 4. Discussion

Recently, Kieffer and Lowndes categorized the DSB repair process into three steps [42]. The ‘immediate-early step’ response is to sense the DSB region when it forms. The ‘early step’ response includes chromatin changes and ATM signaling via the recruitment of key enzymes to protect the damage and prepare for repair. The ‘late step’ response involves the choice of the proper repair pathway depending on the condition. The role of TRIP13 in the DNA damage response has only been reported in the ‘late step’ response, which regulates the choice between HDR and NHEJ [3,9]. However, our results showed that the depletion of TRIP13 does not enhance NHEJ, and that TRIP13 is recruited to a DNA damage site within a few seconds. Moreover, the depletion of TRIP13 inhibits the amplification of DNA damage signaling and recruitment of its downstream signaling proteins, which correspond to an ‘early step’ response. These results suggest that TRIP13 participates not only in the ‘late step’ response, but also in the ‘immediate-early step’ response to DNA damage.

Using a quantitative proteomics approach with a proximity-labeling method, we identified interaction partners of TRIP13 involved in the ‘intermediate-early step’ response to DNA damage. Among several DNA repair-related proteins, MRE11 was selected because the MRN complex initiates DNA strand break sensing and ATM activation [43,44,45]. We confirmed that TRIP13 directly interacts with MRE11 and RAD50, but not with NBS1, upon induction of DNA damage (Figure 2E). Proximity-labeling proteomics and conventional immunoprecipitation did not detect the interaction of TRIP13 and NBS1 (Figure 2E,F); therefore, we speculate that a MRN complex containing only MRE11 and RAD50, but not NBS1, exists. In fact, overexpression of the yeast ATM ortholog TEL1 maintains DSB repair and telomere maintenance without NBS1 [46]. Furthermore, the protein complex database based on a massive co-elution profile (hu.MAP v2) [47] reported a complex containing only MRE11 and RAD50 (HuMAP2_02474; *p*-value 1.98 × 10^−4^), and a complex containing all known MRN complex members (MRE11, NBN (NBS1), RAD50, FAM219A, and HEATR1; HuMAP2_01080; *p*-value 4.96 × 10^−4^). However, further investigation is necessary to confirm the specificity of the interaction between TRIP13 and the MRE11-RAD50 complex.

Interestingly, TRIP13 appears to interact with MRE11 even in the absence of DNA damage. MRE11 was identified as a TRIP13 interaction partner through the SILAC method without DNA damage, and this interaction did not differ in the presence and absence of DNA damage (Figure 2A). Thus, TRIP13 may play a role with MRE11 even in the absence of DNA damage. Another possibility is that TRIP13 interacts with MRE11 for the immediate response to DNA damage. Upon UV irradiation, TRIP13 was recruited at DNA damage sites within a few seconds. Therefore, TRIP13 may interact with MRE11, even in the absence of DNA damage, and rapidly react when spontaneous damage occurs. These hypotheses should be investigated in further research.

Recruitment of MDC1 to DNA damage sites and the physical interaction of MDC1 with the MRN complex were significantly reduced upon TRIP13 depletion. Previously, it was reported that TRIP13 can induce conformational changes of its physical interaction partners through ATP hydrolysis, and two motifs called Walker A (for ATP binding) and Walker B (for ATP hydrolysis) are important for this process [48]. In addition, TRIP13 releases a substrate from a protein complex by inducing a conformational change of the substrate [3,4,49]. However, our results suggest that TRIP13 promotes the interaction between MRE11 and MDC1 rather than inducing the release of MRE11 from the MRN complex or MDC1. It would be interesting to fully elucidate how TRIP13 regulates the interaction between MDC1 and the MRN complex in the future. Collectively, our results confirmed that TRIP13 regulates ATM signaling amplification and affects the interaction between MDC1 and the MRN complex by directly interacting with MRE11.

ATM phosphorylates TQXF motifs in MDC1, creating docking sites for the ubiquitin E3 ligase RNF8 [50], which ubiquitylates histone H1 or L3MBTL2 and initiates recruitment of another E3 ligase, RNF168 [51,52]. Sequentially, RNF168 ubiquitylates histone H2A and triggers the recruitment of DNA damage response factors, such as 53BP1 and its downstream proteins, including RIF1 [53,54,55], PTIP [56], and the Shieldin complex [57]. The Shieldin complex protects DSBs and promotes the NHEJ pathway, and the HORMA domain protein MAD2L2 (Rev7) is part of this complex [58,59]. Previous studies proposed that TRIP13 enhances the HDR pathway after the induction of DSBs by releasing MAD2L2 (Rev7) from the Shieldin complex [3,9]. In these studies, TRIP13 knockdown affected the HDR pathway, rather than NHEJ. Using the I-*SceI*-based GFP-reporter assay, we confirmed that the SSA and HDR pathways were inhibited upon depletion of TRIP13 and NHEJ and alt-NHEJ were slightly affected. It seems that TRIP13 regulates DNA end resection at DSBs to facilitate HDR and SSA, which are required for intermediate- and long-range resection. It is important to note that a subtle change of alt-NHEJ could not be detected in our experimental schedule because the frequency in the alt-NHEJ reporter assay was relatively low, even in wild-type cells.

In conclusion, TRIP13 plays a role in ATM activation by interacting with MRE11 in addition to its role in regulating the choice of the DSB repair pathway. Our results suggest that TRIP13 participates upstream of the ATM signaling pathway corresponding to an ‘immediate-early’ response in DNA damage sensing, enhancing both HDR and NHEJ. Together with the previous results [3,9], our findings demonstrate that TRIP13 is an important player in the regulation of DSB repair at several stages.

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
