# Peer review of "TRIP13 Participates in Immediate-Early Sensing of DNA Strand Breaks and ATM Signaling Amplification through MRE11"

_cells, 2022, doi:10.3390/cells11244095_

Round 1

Reviewer 1 Report

The authors identified MRE11 as a novel TRIP13 partner, and investigated the relevance if this relationship to DNA damage response and DSB repair. They showed that TRIP13 is required for the interaction between MRE11 and MDC1, a mediator of DNA damage response, which promotes MDC1 recruitment to DNA damage sites. They also presented evidence indicating that TRIP13 regulates ATM signaling, thereby aiding in DSB end resection and HR. Their results uncovered novel molecular functions of TRIP13 as an emerging regulator of DNA damage response and repair. The paper is well written and most, if not all, the data are solid and convincing. It would be desirable to include a model figure at the end. 

Minor points:

Lines 82, 83                 change “independent” to “independently”

111                              Table 2 appears before Table 1

Lines 306                     change “independent” to “independently”

Reviewer 2 Report

In this paper  Jeong et al. described a new role of TRIP13 in DNA damage response. In particular the authors found that TRIP13 participates in immediate-early DNA damage sensing and regulates ATM signalling.

The manuscript is  well written. The data presented are convincing and are properly quantified. There are few issues, which should be addressed prior to publication:

-this reviewer suggest to show in figures 1C  that TRIP13 is effectively degraded,  for example through Western Blot analysis. Consistently, also in figure 3D this reviewer suggests to provide a control of downregulation of TRIP13 mRNA in cells expressing RNA interference;

-to support data presented in Figure 1C this reviewer suggests to add a comet assay;

-in figure 3C  this reviewer suggests to add experiments with pS1981 ATM foci

The overall comment on this manuscript it is that the finding is interestingly and new.

Reviewer 3 Report

In this manuscript the authors investigate the role of the protein TRIP13 in the immediate-early sensing of DNA strand breaks. The authors first verify previous research on demonstrate that TRIP13 promotes HDR (homology-directed repair) as well as SSA (single-stranded annealing). The authors also demonstrate that TRIP13 can be seen recruited the DSBs immediately after induction. The authors carefully demonstrate that TRIP13 interacts with MRE11 in both the presence and absence of DNA damage and does not interact with NBS1. The authors show that TRIP31 promotes the recruitment of MDC1 to DSBs by promoting the interaction between MRE11 and MDC1 following IR treatment. Finally the authors also demonstrated that TRIP13 promotes DNA end-resection at DSBs. While some of the results are clear and novel, some seem to be inaccurately or not clearly described. In addition, many of the experimental details are not listed at all, either in the methods or figure legends. This makes it really hard to draw any conclusions from some of the experiments. Below I have gone over in detail any issues.

·       In the previous publication “MAD2L2 dimerization and TRIP13 control shieldin activity in DNA repair” by Krijger et al. in 2021, the authors demonstrate that TRIP13 regulates NHEJ by looking at TRF2-depletion induced end-to-end fusions and CSR. The authors of this paper do not mention this paper at all and actually state that previous research has only looked at the role of TRIP13 in HDR. This will need to be adjusted and addressed in the text.

·       There is a lot of missing information for the I-SceI DSB repair pathway cell lines. There is nothing listed in the materials/methods section for this assay. Were the cells analyzed by FACS? How many cells were analyzed? How many times was the experiment performed and how were the error bars generated? How was significance calculated? It seems that there is a decrease in NHEJ and there is no overlap for some of the error bars for the EJ5-GFP samples and yet there is no significant change. There also does not seem to be any error bars for the EJ2-GFP cells which does not seem right.

·       The technique used for siRNA-mediated depletion should be listed in the methods sections. The sequences of the siRNAs used in this study should also be listed or if used previously those publications can be referenced.

·       Results not clearly described for GFP-TRIP13 recruitment to laser-induced DSBs in the presence of ATPyS. How does it compare to -ATPgS

·       In Figure 3C, it is not described clearly what dose of IR or HU was used to treat the cells for colony survival. HU is not typically used to specifically induce DSBs but rather to induce replication stress, so it is key to list what dose has been used for treatment. CPT would be a better option for DSBs.

·       In Figure 2E, there is no negative control for IP (Strep), so it is hard to say for sure if the interaction with these other proteins is specific to Strep-TRIP13.

·       For IP experiments in Figure 2E and 2F, how long after IR treatment were the samples collected and how long were the cells exposed to 1uM CPT before analysis. This information is key to understanding these IPs. Also, for these IP experiments, it is not described anywhere what DOX treatment is inducing. I am assuming that it is inducing expression of BiolD2-tagged TRIP13, however this is not described anywhere. This should be more clearly stated either in the legend or in the methods.

·       I am concerned about the IP results for Figure 2E and 2F. In 2E, there is interaction between TRIP13 and ATM/RAD50/CtIP, however this interaction is not observed for these proteins in Figure 2F. If there is any difference in the IP protocol that may have caused this, that needs to be clearly addressed.

·       For the IP experiments in Figure 2C-G, was the IP performed identically to that for the proteomics screen. If there are any variations when looking at targeted proteins, there should be a separate Immunoprecipitation section in the methods and materials, especially for the MRE11 IP performed in Figure 2G.

·       In Figure 3A, the authors show that there is a clear increase in chromatin bound MDC1 after IR treatment and this is clearly reduced after TRIP13 depletion. However, the levels of MDC1 in the soluble fraction seems to be very low overall and for the TRIP13 depleted samples there should be a clear increase in the soluble portion after IR treatment, however this is not always the case. It should be tested or somehow demonstrated that depletion of TRIP13 does not alter total MDC1 protein levels in untreated and IR treated samples. It should also be clearly listed at what time post IR treatment the cells were analyzed.

·       For Figure 3C, it should be described either in the methods or the figure legend how the number of MDC1 foci were calculated per cell. Also, how many cells were counted per sample?

·       In Figure 4A, the authors test if TRIP13 regulates the phosphorylation of several ATM kinase substrates following IR treatment. Once again, it is key to describe at what time post IR treatment the cells were collected. Also, the authors state that depletion of TRIP13 downregulates pATM, pKAP1, pS4/8-RPA32 and gH2AX while not affecting pS33-RPA32, however there does not seem to be any visual changes in pATM, pKAP1, pS33-RPA32 or gH2AX levels following IAA-induced depletion of TRIP13. In addition, there does seem to be a decrease in pS4/8-RPA32 following TRIP13 depletion which would agree with the resection data observed in the ER-AsiSI system. This is very concerning as the data presented seems to disagree with what has been writing in text in the results section. Either a different blot needs to be included which more clearly shows that authors message or the level of phosphorylation for each protein needs to be carefully quantified from the western blots compared to the total level of each protein. If this issue cannot be cleared up, the ATM kinase pathway data should be removed as it is not necessary to describe the role of TRIP13 in end-resection, just keep the pS4/8-RPA32 data.

·       Once again, in Figure 4C there is no information about how the experiment was performed for SCE. There is no section in the methods or any details in the figure legend. If the technique was adapted from a previous publication, then this should be referenced. It should also be clarified in these samples are untreated or following any type of damage such as IR.

·       In Figure 4D and 4E, the authors state in the results that they use the “ER-AsiSI GFP-reporter system,” however I have never seen these cells used for the GFP reporter assays and there is no reference listed. Usually, you would use the cell lines that the authors used in Figure 1A, the stable U2OS DR-GFP and SA-GFP, to measure HDR and SA respectively. Is this just a mistake, or were the ER-AsiSI cells used for the GFP assay?

While there is a lot of descriptive information missing in this manuscript, the authors have uncovered a novel interaction between TRIP13 and MRE11 and that TRIP13 regulates the recruitment of MDC1 to DSBs by regulating the interaction between MRE11 and MDC1 following damage. High TRIP13 expression levels have also recently been shown to correlate with poor prognosis in BRCA1-deficient breast cancer cells, so clearly there is a significance to understanding the function of TRIP13 in DSB repair.

Spelling errors

·       In Figure 1B, the panels are labeled ATPrS, should this be ATPgS?

·       In figure 4A, one of the western blot panels is labeled “rH2AX”, should this be gH2AX?

Round 2

Reviewer 3 Report

The authors have sufficient addressed the comments that I had for the first version of the manuscript. I just have one quick comment seen below. Once this is addressed the manuscript is acceptable for publication.

In figure 2B, I am confused why the authors switched the data for this panel. I thought the original data was fine for this panel. The new data is not looking at interaction with MRE11 at all but looking at other proteins (p31comet, MAD2, CDC20). I am assuming this was an error that can be fixed.

Round 3

Reviewer 3 Report

My comments for the modifed Figure 2 were address. My issue was not including the new data, just that the text and figure legend did not line up. This issue was addressed in the newest version.